**Data Availability Statement:** The replication materials are available at https://doi.org/10.7910/DVN/O9NR33.

**Funding:** Miroslav Nemčok acknowledges funding from the Research Council of Norway for the

# Reassessing the winner-loser gap in satisfaction with democracy

**Jean-François Daoust[1,2], Miroslav Nemčok[3]\*, Philipp Broniecki[3], Peter J. Loewen[4]**

1 School of Applied Politics, Université de Sherbrooke, Sherbrooke, Québec, Canada, 2 School of Social and Political Science, University of Edinburgh, Edinburgh, United Kingdom, 3 Department of Political Science, University of Oslo, Oslo, Norway, 4 Department of Government, College of Arts and Sciences, Cornell University, Ithaca, New York, United States of America

\* miroslav.nemcok@stv.uio.no

## Abstract

Citizens who support a party which enters government are systematically more satisfied with democracy compared to voters who supported a party which ends up in the opposition. This relationship is labelled as the "winner-loser gap," but we lack firm causal evidence of this gap. We provide a causal estimate of the effects of voting for a winning or losing party by leveraging data from surveys fielded *before* and *after* new government formations in three well established democracies (Netherlands, Norway and Iceland) were announced in contexts of very high uncertainty. Using a regression discontinuity design comparing citizens' levels of satisfaction with democracy *just before* and *just after* their electoral status (winner or loser) was revealed, we find that the impact of winning or losing is undistinguishable from zero. We conclude by discussing the implications of our findings.

## Introduction

After an election, citizens who voted for a party included in the post-election government are systematically more satisfied with democracy than those who supported an opposition party. This "winner-loser gap" has been identified across a vast literature on political support and especially satisfaction with democracy, and constitutes one of a few remarkably robust findings in political science [e.g., 1–6]. However, this body of research mostly relies on observational data from cross-sectional surveys, where a party winning government is highly correlated with that party being widely supported in the electorate. Are "winning" citizens more satisfied because their party is in power or because their political preference is widely shared in the electorate? This causal mechanism underlying greater satisfaction is thus not well identified. This is worrisome given the importance of understanding citizens' political support in general, but also because the size of the winner-loser gap can inform us about the quality of a democratic system whereas citizens' political support should ideally not be driven by electoral outcomes [7].

To better isolate the impact of winner and losing on citizens' satisfaction with democracy, we implement an innovative strategy leveraging post-electoral contexts for which the incoming government was announced after a period of high uncertainty regarding the parties forming the next government. More precisely, this strategy requires measuring citizens' satisfaction

WELTRUST project, reference number 301443.
The funders had no role in study design, data
collection and analysis, decision to publish, or
preparation of the manuscript.

**Competing interests:** The authors have declared
that no competing interests exist.

with democracy after the election but before the news about the incoming government are released via media. Considering uncertainty in the government formation process, these individuals do *not* know whether they belong among electoral winners or losers, but they do however know how many of their fellow citizens supported the party they supported and how many chose other alternatives. It also requires respondents surveyed during a second time period, that is, once the information about the incoming coalition is reported in the public discourse making citizens aware of their winner/loser status. If the gap between those who voted for a party ending up in government and those who supported an opposition party is caused by their winner/loser status (as the underlying mechanism of the winner-loser gap implies), we should see an emerging gap at the time the electoral status is known by citizens in a context of high uncertainty. This theoretical timeline is illustrated in Fig 1 below.

Since the timing of the interview is exogenous (we come back to this point later), we can estimate the causal effect of winning or losing an election on citizens' satisfaction with democracy by comparing individuals surveyed *just before* and *just after* [cf. 8]. However, post-electoral contexts where the number of seats allocated to parties are known but with uncertainty about the upcoming government formation are comparatively rare. After a thorough review of the largest comparative datasets, we found three cases satisfying the strict criteria, that is, the parliamentary elections in Norway 1997, the Netherlands 2012, and Iceland 2017.

Surprisingly, our findings do not reveal any discernable effect of voting for a government or opposition party on voters' satisfaction with democracy, and this main conclusion holds across a variety of model specifications (adjusting cutoff identification, array of permutation tests, etc.). One implication is that even though the literature attributes the differences in satisfaction among winners and losers to the parties' access to the government, our findings make it look unlikely that this is the whole story in our sample of well-established consensual democracies (Norway, Netherlands and Iceland). The causal mechanism behind the winner-loser gap appears to be more complex and more research should address this puzzle in several ways that we discuss in the conclusion.

## Satisfaction with democracy and the winner-loser gap

Electoral outcomes, by generating winners and losers, are important sources of support for political systems. However, losers are systematically less likely to express political support

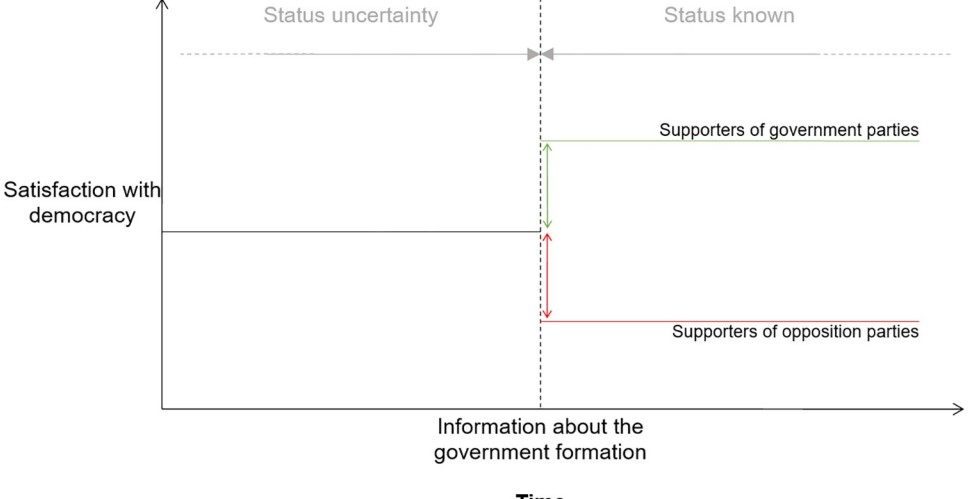

**Fig 1. Theoretical timeline of government formation and the winner-loser gap.**

compared to winners. This is the key puzzle tackled by the classic work of Anderson et al. [1], *Losers' Consent*, which argues that stability and sustainability of democratic systems are dependent on understanding of the *losers'* post-electoral responses because they are consistently revealed to be less satisfied with the way democracy works in their country [9].

Past work suggests different theoretical mechanisms generating the winner-loser gap in satisfaction with democracy, but can be roughly divided in two: emotion- and utility-based mechanisms. First, as is the case in many domains, people prefer winning over losing because it generates positive emotions such as joy and pride [10,11]. Second, citizens can expect more or less utility from the system (through public policies) based on whether the party they supported is part of the government or is in the opposition [3,4]. Indeed, increasing ideological and policy congruence between voters and elected representatives is found to increase the overall level of satisfaction with democracy, and this tendency holds for electoral winners as well as losers [12–15]. Overall, it is however not clear which mechanism(s), if any, drives the gap [16].

In contrast to the uncertainty regarding the mechanism(s), the literature is much clearer when it comes to defining the winners and losers. Being a "winner" or a "loser" taps onto a party's performance. Many measures can be used to capture party performance. First, one can focus on vote share, seat share and/or participation in government [17]. Second, these three sets of indicators can be either based on a given election (absolute performance) or compared to the party's previous electoral results (relative performance) [18]. However, the focus has been put on whether a party is included or excluded from a government, that is, whether one's party is in power or not [1]. Winners are thus voters who supported a party that ends up with cabinet members, while losers are supporters of parties ending up in the opposition. Stiers, Daoust, and Blais [19] examine citizens' own perceptions of having won or lost an election and show that whether a voter's party is included in government is the most important determinant of whether the voter views themselves as a winner. In other words, being in government *is* winning. That said, other measures like the relative vote share between the previous election and a given election is also useful, and the media's coverage of the electoral outcomes also shape which parties are depicted as winners or losers [20,21]. At the end of the day, however, we follow the conventional wisdom, and in particular the pioneering work of Anderson et al. [1], by defining winners and losers based on whether the party they supported is included or excluded from the government.

Attempts to isolate the causal effect of winning/losing faces serious challenges. Moving beyond cross-sectional data, panel studies surveying respondents before and after election constitute an important step forward by focusing on within-individual differences over time [5,16,22–25]. For example, Erhart recently leveraged Dutch panel data to examine how various indicators of representation (e.g., party closeness) shape citizens' political support [26]. One alternative strategy is to measure satisfaction within an experimental context, in which winning or losing is randomly assigned. But this is not an option to study real-life elections. One solution is to search for an empirical context in which most citizens cannot be confident in knowing whether their party has yet won or lost the election, but nonetheless know how much support their party garnered in the election.

Halliez and Thornton [27] followed this approach focusing on the 2000 US presidential election and the reinterviews in 2002 suggest that, in the absence of a clear election outcome, future winners and losers tend to reveal comparable levels of satisfaction with democracy and the winner-loser gap forms only once the uncertainty is resolved. Although the approach is original and insightful, there are two key limits in their study. First, it focuses on an unusual case from a single democracy, i.e., the US. Second, and more importantly, their data do not allow a comparison of voters *just before and after* the electoral status is known given that they

use the post-electoral survey of 2000 compared to attitudes expression two years later during the midterms of 2002. In this research, we focus on three cases where we were able to provide such comparisons.

In taking up this research design, we aim to separate two distinct (but not mutually exclusive) mechanisms for the winner-loser gaps. The first mechanism, which we have noted above is sometimes regarded as an "emotional" mechanism, is that in which voters who vote for more popular parties are more satisfied precisely because that party is more popular. Being onside with a large share of other citizens and perhaps knowing that they share one's views may be sufficient to generate more satisfaction with democracy. This can be labelled as a "popularity mechanism". Note that this emotion-based mechanism is different than the hypothesis suggesting that people are more satisfied with democracy when they feel that they "won" the election [16].

But because parties which earn a greater share of the vote are, *ceteris paribus*, more likely to be in government, this may conflate with the status of having voted for a party included in the government, which is a different mechanism. This second mechanism involves having the party you support actually be in government. Having one's preferred party in government can confer both symbolic benefits and material ones, and both can be expected to increase satisfaction with the democracy. This can be called a "government mechanism". In this research, we aim to isolate this very mechanism.

## Methodology

Our research design requires post-electoral circumstances in which the composition of the incoming government was highly uncertain. In such context, voters are not aware of their winner/loser status before the new government is announced, but they do know how much support their party garnered in the election. After the announcement, the effect of newly acquired winner/loser status kicks in and we can measure the magnitude of its effect by comparing the two groups–those interviewed *just before* (i.e., non-treated) the government formation who is unaware of its status, and those interviewed *just after* (i.e., the treated) the government formation who is aware of its status. See Fig 1 for an illustration of this theoretical timeline.

Moreover, we assume that the timing of the survey interview is exogenous to the preferences of respondents. Indeed, while it is plausible that respondents can influence interview timing, it is not plausible that they do so strategically. Hence, respondents who were surveyed slightly before the government formation date should be almost identical, at least in all relevant aspects, to those who were surveyed slightly after the information about the new government became known. This expectation is confirmed by the data: as shown in Appendix B in S1 Appendix, there is systematically no differences between the means of our variables before and after the treatment, with few exceptions for which the differences are not substantially important (e.g., mean age of 56 versus 52 in the Netherlands). In other words, the cutoff date, that is, when the government formation became known, is treated 'as if' random' [28].

## Data

To find post-electoral contexts fitting the criteria detailed above, we surveyed two of the largest comparative datasets including key indicators, that is, the Comparative Study of the Electoral System (CSES) and the European Social Survey (ESS). Altogether, they include 389 surveys (195 in CSES and 194 in ESS) conducted in 57 countries during more than two decades (1996–2018). These relevant organizations confirm adherence to ethical standards, relevant legal regulations, and the provision of informed consent for all included participants. Among the available surveys, we identified cases with uncertain post-electoral development based on

three rounds of exclusion. In the first round, we exclude cases due to data limitation. The most obvious criterion related to data availability is whether a post-electoral survey was in the field before *and* after the incoming government was announced in public discourse. The cases where observations are not available on both sides of cutoff were excluded. We also removed elections where the mode of data collection was a "mail back survey" such as in New Zealand, due to uncertainty around the timing the survey completion. In a second step, we excluded cases without uncertainty regarding voters' winner/loser status. Hence, we removed elections in which a party received legislative majority, and where formation of the new government repeated the pattern from previous electoral cycle as it is reasonable to expect that citizens may anticipate such coalition due to the recent experience.

In the third step, we conducted a qualitative inquiry into the remaining elections by reviewing national election reports and news media to determine the level of uncertainty related to the government formation process. At the end of the day, three elections that satisfied our scope conditions were: Norway (1997) and Iceland (2017) from the CSES and the Netherlands (2012) from the ESS. For descriptive statistics relevant to the winner-loser gap and the timing of survey interviews, refer to S-A1 Fig in Appendix A in S1 Appendix. For additional information on the political context surrounding the three cases, see Harðarson and Önnudóttir [29], Van Holsteyn [30], Madeley [31], Pellikaan, de Lange, and van der Meer [32], and Valen [33] for electoral reports. The external validity of our case studies is constrained by the fact that the three countries included are not representative of most democracies in the world. Indeed, they are well established democracies using proportional representation, which has been shown to reduce the winner-loser gap [34]. That said, even democracies of highest quality still display a winner-loser gap in citizens' political support [35]. Indeed, Erhart mentions that "The winner-loser gap persists even in a consolidated, well-functioning consensus democracy like the Netherlands." [26] However, given the characteristics of the countries included in our samples, we do not extrapolate our findings to democracies in general, but rather to the three case studies that we could include.

## Measures

The survey question measuring satisfaction with democracy in Norway and Iceland was "On the whole, are you very satisfied, fairly satisfied, not very satisfied, or not at all satisfied with the way democracy works in [*country*]?" Netherlands used a slightly different wording: "And on the whole, how satisfied are you with the way democracy works in Netherlands?" with answer choices ranging from 0 (extremely dissatisfied) to 10 (extremely satisfied). We normalized the scores across the three datasets, centering the distribution at 0. As a single item measure, the question aims to capture people's satisfaction with the performance of democratic institutions in practice, rather than support for democratic principles or the legitimacy of a democracy regime [36,37].

The winner-loser status was coded based on respondents' self-reported vote choice. After the 1997 elections in Norway, the government was formed by the Liberal Party, the Christian People's Party, and the Center Party. Hence, their self-reported voters are coded as winners, while all other responses referring to Norwegian political parties are coded as losers. In Iceland, following the 2017 elections, the post-election government was formed by the Progressive Party, the Independence Party, and the Left-Green Movement. Thus, their voters are considered winners in our analysis, while respondents who chose other parties are coded as losers. In the Netherlands, after the 2012 elections, a minority cabinet was formed by the Party for Freedom and Democracy and the Labour Party, with support from the PVV (List Wilders). Consistent with Erhard [26], we coded only the voters of the first two parties as winners, as

these were the only parties with representatives nominated into the government. Since the PVV did not have a government portfolio and did not directly participate in the government, their voters were coded as losers, along with voters of other opposition parties. In all three cases, abstainers were excluded from the analysis, but we should note that merging them with losers do not alter our conclusions, see S-D1 Table in Appendix D in S1 Appendix. The full list of party choices and their corresponding coding is provided in S-A1 Table in Appendix A in S1 Appendix.

On the use of recall question regarding respondents' vote choice and their validity, see Fournier et al. [38] and Dassonneville and Hooghe [39]. We followed the conventional wisdom regarding the notions of winning and losing (as reviewed previously) by conceiving as winners voters who cast their ballot for a party included in the governing coalition after the election. Losers are those who supported an opposition party. To be considered as "included" in government, a party needs to have at least one appointed representative in the cabinet. Hence, parties who support a government in parliament (whether informally or formally by following, for example, a confidence and supply agreement) but do not have any representation through a cabinet member are considered as not being part of the government. This approach assumes that being in or out of the government is key [1]. This assumption is, we believe, reasonable (which is evidenced by the scholarship using this approach), and also echoes Daoust, Plescia and Blais' [16] findings on citizens' public opinion: "Yes, having voted for a party making it to the government produces a feeling of winning, and yes, it leads to greater satisfaction with the state of democracy."

Even though all three cases embody proportional representation systems [40], they nevertheless constitute distinctive contexts in terms of the winner-loser gap in satisfaction with democracy. We conduct a regression of satisfaction with democracy on whether individuals are winners or losers, using only the survey interviews conducted following the announcement of the new government. The resulting coefficient plot is presented in Fig 2. The estimated winner-loser gap in satisfaction with democracy is insignificant in Norway, moderately large in the Netherlands (approximately 0.2 standard deviations), and quite substantial in Iceland (exceeding 0.5 standard deviations). Therefore, despite similar institutional setups, the

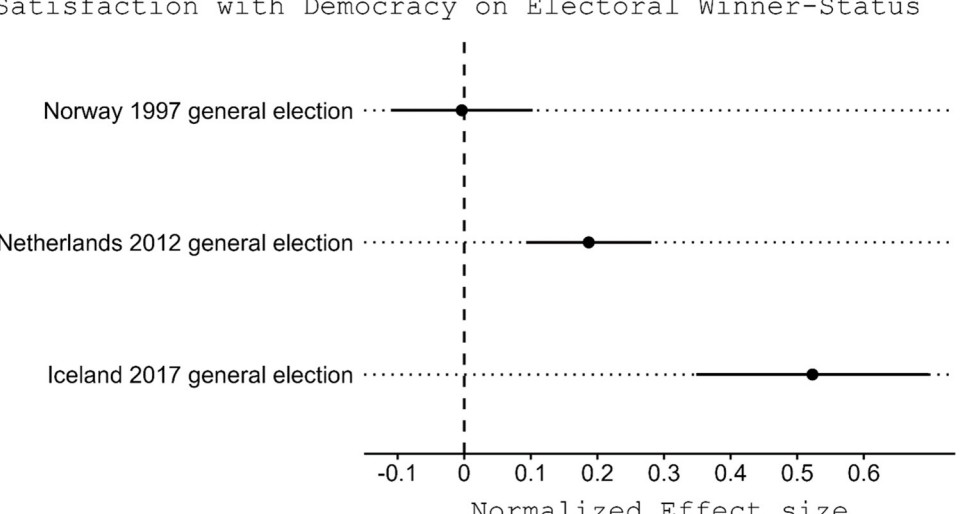

**Fig 2. Standard regression models: The winner-loser gaps in satisfaction with democracy in Norway (1997), the Netherlands (2012), and Iceland (2017).** The horizontal lines represent 95% confidence intervals. The regression uses only the interviews conducted after the announcement of the new government.

contextual conditions of the three cases differ, allowing the analysis to address the effect of changes in winner/loser status change across three contexts that vary in the magnitude of the winner-loser gap.

## Findings

Before presenting the findings from the regression discontinuity analysis, we present the distributions of the interviews on the timeline around the date when the winner/loser status became known, as well as the distribution of winners and losers in the samples. Fig 3 provides this information pooling all three elections (see Appendix A and S-A1 Fig in S1 Appendix for details by election), including a window of a month around the actual date when the electoral status was revealed.

In our analysis, we follow the now canonical approach of automatic bandwidth selection with variable bandwidths on both sides of the cutoff [41]. We accept the default settings of the *rdrobust* (v2.2) package for the statistical software R [42] with a uniform kernel to increase power.

The package estimates two bandwidths determined by the data following the Calonico, Cattaneo, and Farrell's approach [41]. The first bandwidth is used for the conventional RD estimates and the second for the bias-corrected estimates [41,43]. The results are presented in Table 1 based on the pooled sample, while separate analysis for each country are shown in Appendix C in S1 Appendix. We estimate satisfaction with democracy at and around the cutoff following the standard approach via local polynomial regression with a first order polynomial to avoid overfitting [44,45].

The conventional RD estimates are very well powered. The samples include a substantial amount of variation with numerous respondents just before and just after the cutoff date. The effect for the electoral winners is significant but in the wrong direction. Electoral winners should be more satisfied with the way democracy works once they learn their electoral status. The effect is insignificant for the electoral losers. For the bias-corrected estimates, the design has a power of 0.84 for electoral winners at an effect size of 0.25. For electoral losers the effect size would need to be around 0.5 to have sufficient power of 0.84 with bias-corrected estimates. These effect sizes are reasonable and stable for the conventional, bias-corrected and robust methods. Thus, we conclude that we have sufficient power for our tests.

The exact window sizes and regression discontinuity estimates for each case separately are shown Appendix C and S-C1 to S-C3 Tables in S1 Appendix.

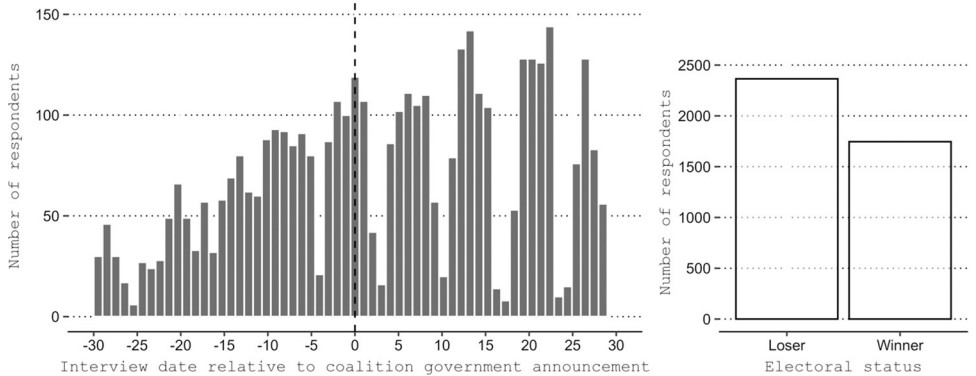

**Fig 3. The timeline of the responses and the electoral status.** All elections pooled. The total number of respondents is 4,118.

**Table 1. Pooled elections RD estimates.**

| | Electoral Winners | | | Electoral Losers | | |
|---|---|---|---|---|---|---|
| Method | Coefficient | 95% CI | | Coefficient | 95% CI | |
| Conventional | -0.30 | -0.58 | -0.03 | 0.05 | -0.26 | 0.36 |
| Bias-Corrected | -0.32 | -0.59 | -0.04 | 0.04 | -0.27 | 0.34 |
| Robust | -0.32 | -0.61 | -0.02 | 0.04 | -0.29 | 0.36 |
| N | 2037 | | | 2628 | | |
| BW est. (std.) | 6/16 | | | 6/19 | | |
| BW est. (robust) | 19/59 | | | 16/74 | | |

Estimated with the rdrobust package (v2.2) in R. Reported methods are 1) conventional RD estimates with conventional standard errors, 2) bias-corrected estimates with conventional standard errors, 3) bias corrected estimates with bias-corrected standard errors.

The RD findings are also visualised as Fig 4 to facilitate a more intuitive interpretation of the temporal dynamics of the satisfaction with democracy. First, the left panels show the regression discontinuity plots for both winners and losers. We compute both conventional and robust RD estimates which apply automatic bandwidth selection. As already demonstrated in Table 1, the results are substantially similar regardless of the bandwidth choice. Second, the right panels present permutation analyses. This test splits the data into a pre-uncertainty and a post-uncertainty period 10,000 times, and then randomly selects a cutoff. We then estimate a linear fixed-effects model and keep the difference in satisfaction with democracy between the pre and post periods. Finally, we compare the distribution of differences (difference in means) with the actual difference in means (i.e., when the cutoff is 0).

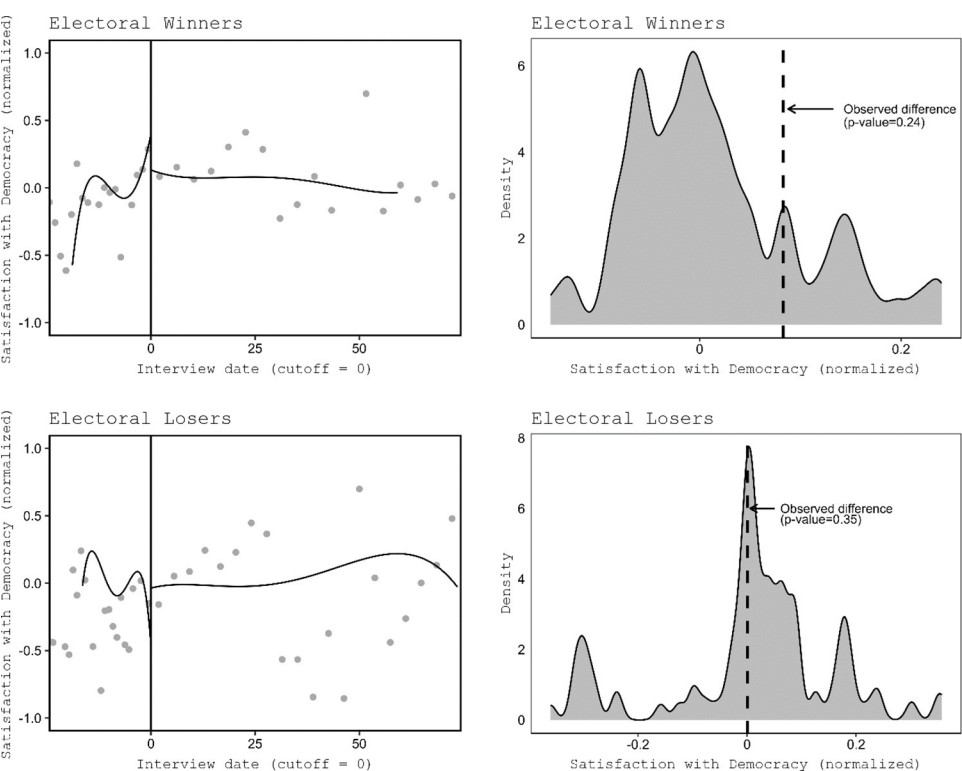

**Fig 4. Satisfaction with democracy and the post-electoral government formation.** First row: Electoral winners. Second row: Electoral losers. First column: RDD plots. Second column: Permutation tests.

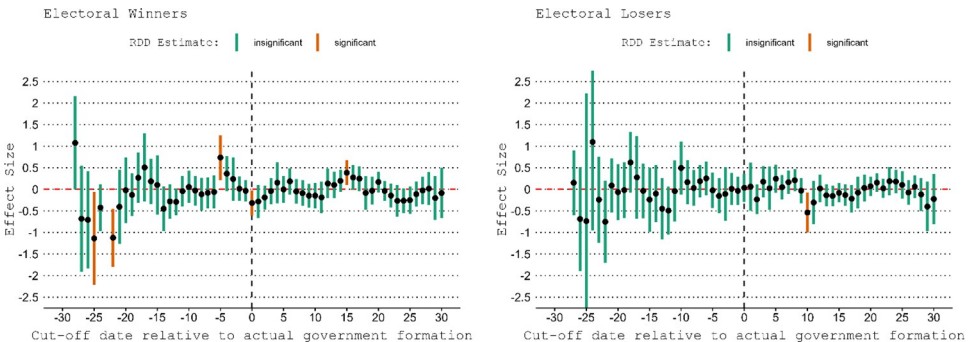

**Fig 5. All elections pooled. RDD's with varying cutoffs.** Robust Doughnut RDD estimates with varying cutoff dates on the x-axis and 95% confidence intervals.

Our results suggest that there is no consistent effect of the electoral status (that is, whether a citizen voted for a party ending up in governing versus in the opposition) on satisfaction with democracy. Based on the huge literature on the winner-loser gap, we expected an increase at the cutoff for winners, as well as a decrease for losers. However, we do not see such patterns as the effect on electoral winners is negative and the effect on electoral losers is not distinct from zero. The impact on satisfaction with democracy is estimated to be -0.32 (95% confidence interval: -0.61 to -0.02) for winners and the effect of losers fails to reach conventional levels of statistical significance (respectively $p = 0.83$).

The upper left panel in Fig 4 showing that satisfaction with democracy among winners seems to increase some 20 days before the cut-off could be consistent with reasonable hypothesis that voters may anticipate the news about the government formation even before they are published by media in response to which their satisfaction with democracy increases. This would be problematic for our identification strategy: respondents could find out about their winner/loser status slightly before the status become known due to rumors. In fact, it is also plausible that respondents found out about their electoral status slightly after the incoming government is publicly announced via media because people do not always follow the news closely enough. We test these possibilities by varying the cutoffs along the time dimension. For each day until 30 days before and 30 days after the actual cutoff, we estimate separate RDD's. One caveat to this type of placebo analysis is that we expect to see about 3 false positives per plot because we estimate 95% confidence intervals and estimate 60 models. Fig 5 shows the results for the pooled models (while S-D1 Fig in Appendix D in S1 Appendix shows the effects for each separate election). The findings are not consistent with the anticipation hypothesis. The left panel of Fig 5 shows the results, and one would expect from the anticipation mechanism that winners should become more satisfied with the way democracy works once they find out about their electoral status. There are only two positive and significant effects (out of 60 estimates) in the left panel (winners only): One five days before the actual cutoff and one fifteen days after. Both of these effects could be false positive effects. The right panel illustrates the effects for the electoral losers. Electoral losers should become less satisfied once they learn their electoral status. Here, we therefore expect a negative effect. There is one significant negative effect in the plot at ten days after the actual cutoff date. Overall, across both plots there are three effects (out of 120 estimates) that are significant and in the expected direction of the anticipation mechanism. However, the dates do not overlap for winners and losers which goes against the anticipation hypothesis. Indeed, we would expect both winners and losers to learn about their electoral status at about the same day (not 15 days apart—from the positive effect among winners at day -5 to the negative effect among losers at day +10).

All in all, our main finding, although only applying to a subsample of well established democracies, is inconsistent with the literature: knowing whether one voted for a party ending up in government or in the opposition does not seem to have an independent impact on satisfaction with democracy. The results are robust to different specifications such as the selection of the bandwidth, for the elections analysed individually as well as pooled, and when varying the cutoff dates daily until 30 days before and 30 days after the election (see Appendix C and Appendix D in S1 Appendix).

## Discussion

The winner-loser gap is one of the key factors in the study of political support and democratic attitudes [1,6]. However, there has been little work aiming to isolate the causal effect of winning government from the effect of supporting a party along with many other citizens. In this research, we implemented an original research design leveraging a RDD approach and permutation tests that relied on the fact that in some instances, the post-electoral context provided citizens very little clue about the composition of the incoming government – and thus, whether they are winners or losers according to the classical definition. In that context of uncertainty, voters cannot know confidently whether they won or lost the election [27]. In the search for suitable cases, we applied strict criteria and managed to find three clean cases study from Norway (1997), the Netherlands (2012) and Iceland (2017).

Our results do not reveal any satisfaction boost among winners, nor satisfaction decay among losers. Hence, we do not replicate what is systematically found in social science research [e.g., 1–5]. This is puzzling, especially given the robustness and consistency of our key finding from three different contexts and several alternative methodological choices. One of the key implications of our research is that the very act of parties entering government which translates into one's winner/loser status according to the literature seems to be a mere proxy in explaining the winner-loser gap. In other words, party government status does *not* seem to explain why winners tend to be more satisfied with democracy than losers. However, our findings are limited to a set of well-established democracy using proportional representation. Further research should aim to expand cases study outside well-established democracies to assess the external validity of our findings, and should also aim at testing contextual effects related to whether the winners supported a major or junior coalition member of the government, as well as levels of polarisation of a party systems.

## Supporting information

**S1 Appendix.**
(PDF)

## Author Contributions

**Conceptualization:** Jean-François Daoust, Miroslav Nemčok, Philipp Broniecki, Peter J. Loewen.

**Data curation:** Jean-François Daoust, Miroslav Nemčok, Philipp Broniecki.

**Formal analysis:** Jean-François Daoust, Miroslav Nemčok, Philipp Broniecki.

**Investigation:** Jean-François Daoust, Miroslav Nemčok, Philipp Broniecki, Peter J. Loewen.

**Methodology:** Jean-François Daoust, Miroslav Nemčok, Philipp Broniecki.

**Project administration:** Jean-François Daoust.

**Resources:** Jean-François Daoust.

**Validation:** Jean-François Daoust, Miroslav Nemčok, Philipp Broniecki.

**Visualization:** Jean-François Daoust, Miroslav Nemčok, Philipp Broniecki.

**Writing – original draft:** Jean-François Daoust, Miroslav Nemčok, Philipp Broniecki, Peter J. Loewen.

**Writing – review & editing:** Jean-François Daoust, Miroslav Nemčok, Philipp Broniecki, Peter J. Loewen.

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
