## [Decision Letter · Decision Letter 0]

6 May 2024

PONE-D-24-06817Reassessing the Winner-Loser Gap in Satisfaction with DemocracyPLOS ONE

Dear Dr. Nemcok,

Thank you for submitting your manuscript to PLOS ONE. After careful consideration, we feel that it has merit but does not fully meet PLOS ONE’s publication criteria as it currently stands. Therefore, we invite you to submit a revised version of the manuscript that addresses the points raised during the review process. Please submit your revised manuscript by Jun 20 2024 11:59PM. If you will need more time than this to complete your revisions, please reply to this message or contact the journal office at plosone@plos.org. Please include the following items when submitting your revised manuscript:A rebuttal letter that responds to each point raised by the academic editor and reviewer(s). You should upload this letter as a separate file labeled 'Response to Reviewers'.A marked-up copy of your manuscript that highlights changes made to the original version. You should upload this as a separate file labeled 'Revised Manuscript with Track Changes'.An unmarked version of your revised paper without tracked changes. You should upload this as a separate file labeled 'Manuscript'.

We look forward to receiving your revised manuscript.

Kind regards,

Ivan Filipe de Almeida Lopes Fernandes, Ph.D.

Academic Editor

PLOS ONE

Journal Requirements:

"Miroslav Nemčok acknowledges funding from the Research Council of Norway for the WELTRUST project, reference number 301443."

**Additional Editor Comments:**

This manuscript employs a regression discontinuity design (RDD) to assess the impact of political winner/loser status on democratic satisfaction. The causal identification strategy relies on pinpointing public opinion surveys conducted concurrently with the announcement of government formation outcomes. The sample comprises three elections from Western Europe, specifically in the Netherlands, Norway, and Iceland. Nevertheless, several concerns raised by the referees must be addressed in a revised version of the manuscript.

It is necessary to review the mistaken logic of the Balance Tests presented in the appendix, which are crucial for the validity of the research design. It is necessary to discuss the limits of the external validity of the test with the analysis of only 3 elections in countries that have many similar characteristics, which are far removed from the reality of most contemporary democracies. Thus, despite the interest of the findings, which need to be validated with more tests, their conclusions need to be appropriate to the limits of the research design adopted. We kindly ask the authors to pay attention to all the reviewers' comments.

Reviewers' comments:

Reviewer's Responses to Questions

**Comments to the Author**

1. Is the manuscript technically sound, and do the data support the conclusions?

Reviewer #1: Yes

Reviewer #2: Partly

Reviewer #3: Yes

2. Has the statistical analysis been performed appropriately and rigorously? 

Reviewer #1: Yes

Reviewer #2: Yes

Reviewer #3: Yes

3. Have the authors made all data underlying the findings in their manuscript fully available?

Reviewer #1: Yes

Reviewer #2: Yes

Reviewer #3: Yes

4. Is the manuscript presented in an intelligible fashion and written in standard English?

Reviewer #1: Yes

Reviewer #2: Yes

Reviewer #3: Yes

5. Review Comments to the Author

Reviewer #1: Review of ‘Reassessing the Winner-Loser Gap’

This manuscript uses a regression discontinuity design (RDD) to estimate the effect of political winner/loser status on satisfaction in democracy. The causal identification strategy is based on identifying public opinion surveys that took place while the outcomes of government formation processes were announced. The sample includes three elections that took place in Western Europe (the Netherlands, Norway and Iceland).

As the authors rightly note, the gap in satisfaction with democracy and other attitudes between political winners and losers is one of the best documented and robust findings of empirical pollical science. Tracing the mechanisms that account for this gap is an important problem in this literature. In principle, using RDD based on the quasi-random assignment of survey respondents to winner/loser status based on the timing of announcing the outcomes of government formation processes can be a valid strategy. The empirical basis of this paper, however, is quite limited with 3 elections that all took place in old, well-established democracies with proportional representation – all factors that attenuate the effects of electoral winning and losing. Moreover, the government formation process in the Netherlands in particular ended in an outcome that makes it very hard to assign winners and losers, even in the narrower sense of government participation, since the PVV ended up as a party supporting the government but not formally being part of it. (In addition, CDA became part of the government but suffered a significant loss at the election and lost the prime-ministerial position, which makes it a very unusual type of ‘winner’).

So, while the result that using this particular design on these three elections, one cannot detect an effect of political winning on satisfaction with democracy is interesting and valid (even taking into account some questions on the methodology that I have below), I think the relevance of the result is limited based on the empirical scope of the analysis.

Balance tests

A crucial assumption underlying the research design of the paper is that the assignment of survey respondents before and after the announcement of the new government is as-random. That would be the case if the survey recruitment procedure was truly random and there was no differential non-response to the survey that was related the winner/loser status. To validate this assumption, the authors present balance tests in the Appendix. These balance tests, however, examine whether the populations of winners and losers are different before and after the cut-off, which is not the question of interest. The question of interest is whether the samples of respondents interviewed before and after the cut-off point differ across a range of possibly-relevant characterises. This would not be the case, for example, if stratified random sampling was employed by the survey companies collecting the data, where some regions of the country interviewed before others. The design assumption would be violated as well if political losers had a higher propensity to decline an interview on a (political) survey after the announcement of the elections (or to reveal who they voted for).

Another crucial assumption on which the design rests is that the news about the government formation could not have been anticipated and internalized before the announcement. I think this is hard to justify in some contexts, where potential coalition partners strike a deal on general participation in government together before hammering out over months the details of the coalition agreement (such as the Netherlands). In such cases, one would not indeed expect that the announcement of the government formation itself changes much for anyone.

On a side-note, to compare how their estimates differ from panel-data based estimations of the winner-loser gap, the authors could use the LISS survey in the Netherlands and estimate the same effect with that alternative data source and design, in order to be able to pinpoint more precisely where the difference with existing findings comes from.

Additional points

The authors could make it clearer that while participation in government is one, and perhaps the most-commonly used indicator of winning or losing in politics, it is definitely not the only indicator and there is active research going one that tries to establish what makes someone (feel as) a political winner/loser and the role of media in that.

The data is consistent (cf. Figure 4) with the announcement of the government increasing satisfaction with democracy both for winners and losers in the 20 or so day after the cut-off (and in the few days before the cut-off, which is consistent with the anticipation of the news of government formation I mentioned above). I think this is an interesting result that, if confirmed by formal tests, should be mentioned.

The conclusions of the article can do more to unpack the significance of the results and consider the limitations imposed by the selection of countries, elections, and government formation outcomes (and shorten the summary of findings in that section).

Reviewer #2: The manuscript has employed an interesting empirical and methodological strategy to strengthen the causal evaluation of the theory of the winner-loser gap, yielding an unexpected result. However, I believe that the manuscript could be enhanced both in terms of theoretical framing and conclusion formulation, as well as in its empirical analyses. Firstly, in my understanding, the quasi-experimental approach employed by the authors should be regarded as a tool to ascertain which of the two mechanisms proposed to explain the winner-loser gap—emotional or utility-based—is more appropriate or robust, rather than testing the theory as a whole. The manuscript itself presents reasoning akin to this when it criticizes studies based on observational data: “Fundamentally, however, these models cannot disentangle the effects of the positive emotional effect of having voted for a widely supported party from the effect of having voted for a party that controls the government” (rows 136–8).

Therefore, it could be argued that even if the difference in the degree of satisfaction with democracy between winners and losers is attributed to differences in party support in the electorate (the omitted variable that would cause bias, according to the authors), the theory itself remains valid. However, the specific mechanism underlying it is linked to the positive and negative emotions engendered by being in the majority or minority in society.

In this sense, given that the manuscript’s methodology allows for circumventing the previous knowledge of the winner-loser position, I believe that the authors actually test the occurrence of the utility mechanism, which is related to political system outcomes. Therefore, I suggest that the authors present data related to party and electorate polarization in the analyzed elections. If these statistics indicate weak polarization, the null results found could be interpreted as demonstrating that, in the cases examined, weak party and electorate polarization explain the null result, suggesting that it may be the more important mechanism for explaining the occurrence of the winner-loser gap.

Secondly, I believe one empirical aspect that could be important to incorporate into the analyses is the segregation of the winner group into supporters of majority parties and supporters of minority parties. It could be argued that the impact on someone's satisfaction with democracy, based on whether the party they voted for is or is not in government, varies according to the size of the party. In the analyzed cases, this aspect is even more pressing considering that, as I understand it, the manuscript indicates that there was uncertainty regarding the upcoming government formation at the time of public opinion collection, but the number of seats allocated to parties was known (row 87). Theoretically, this could be linked to the comparison of whether and how the winner-loser gap occurs and varies within and between parliamentary and presidential systems, as well as in coalition and single-party governments. Empirically, this could help explain the unexpected negative coefficient found among winners.

To finish, one minor point. In rows 102–3, the manuscript says, “However, losers are systematically less likely to express political support compared to losers.” I think the second “losers” should be replaced by "winners.”.

Reviewer #3: General assessment:

The paper addresses whether the winner-loser status explains satisfaction with democracy. By applying a regression discontinuity design to data from three elections—Norway 1997, the Netherlands 2010, and Iceland 2017—the authors’ main finding do not confirm the theoretical expectations. In other words, winners do not show higher levels of satisfaction with democracy, and losers are not less satisfied with democracy. Although limited to three cases, the causal inference design allows for their main claim: the winner-loser gap does not explain satisfaction with democracy, challenging decades of knowledge regarding the topic.

Whereas the paper’s main strength is establishing a causal relation between the winner-loser status and satisfaction with democracy, its weakness might be the impossibility of external validity. On the one hand, the authors address the question through a careful research design that considers the assumptions of RDD and assigns robustness to their findings through standard tests. On the other hand, it is hard to claim that such findings are valid for other cases rather than those analyzed.

Major issues:

The paper has no significant issues.

Minor issues:

The way in which the authors present their conclusion could be done more carefully. Both in the introduction (lines 97-98) and conclusion (lines 303-306 and 310-311), the authors state their conclusion in a way that might make it seem as if previous literature is misleading - which is not accurate. Although prior tests usually rely on probabilistic research designs (therefore, establishing a correlation between variables) using the winner-loser gap as one of the predictors of various dependent variables outcomes (satisfaction with democracy, trust in political institutions, etc.), the study of three cases alone does not falsify all previous research. The authors are not explicitly arguing this, but the way that they present it – that “party government status does not seem to explain why winners tend to be more satisfied with democracy than losers” – sounds like the findings are generalizable.

Approaching the issue through causal inference techniques is undoubtedly an important step. However, do the conclusions apply beyond these cases? Is there external validity? The findings bring some theoretical implications because theoretical expectations are not confirmed. Yet, the authors should be more cautious in presenting their argument, clarifying that the conclusion only applies to the cases under scrutiny.

The minor issue mentioned above is the most crucial one. Below, I point to a few less substantial but important things to address.

The article needs to be revised for typos and misplaced sentences. A few sentences appear in the main text but should be footnotes or endnotes (lines 197-200 and 212-214, for instance).

Regarding how authors present their findings, I suggest bringing Table 4 from the Appendix to the text.

Figure 4 presents the results. I believe the authors are plotting polynomial regressions, which is a valid method. However, I wonder if a simple linear fit would be better for visualizing continuity assumptions. Although the authors do not confirm expectations, they find significant results contrary to the expectations for winners. Therefore, the running variable discontinuously jumps at the cutoff—even if it jumps downwards. Would the jump occur in the absence of the treatment?

Still related to Figure 4, the authors could centralize the cutoff on the plot. The X-axis should have the same number of days on each side of the cutoff.

Overall, this is a fascinating and promising paper that applies cutting-edge methods to a classic question in public opinion studies. Therefore, my recommendation is to revise minor issues and resubmit.

6. PLOS authors have the option to publish the peer review history of their article (what does this mean?). If published, this will include your full peer review and any attached files.

Reviewer #1: No

Reviewer #2: No

Reviewer #3: No

---

## [Author Response · Author response to Decision Letter 0]

3 Sep 2024

Response to reviewers is provided in the attached document.

---

## [Decision Letter · Decision Letter 1]

20 Nov 2024

Reassessing the Winner-Loser Gap in Satisfaction with Democracy

PONE-D-24-06817R1

Dear Dr. Nemcok,

We’re pleased to inform you that your manuscript has been judged scientifically suitable for publication and will be formally accepted for publication once it meets all outstanding technical requirements.

Kind regards,

Ivan Filipe de Almeida Lopes Fernandes, Ph.D.

Academic Editor

PLOS ONE

Additional Editor Comments (optional):

Reviewers' comments:

Reviewer's Responses to Questions

**Comments to the Author**

1. If the authors have adequately addressed your comments raised in a previous round of review and you feel that this manuscript is now acceptable for publication, you may indicate that here to bypass the “Comments to the Author” section, enter your conflict of interest statement in the “Confidential to Editor” section, and submit your "Accept" recommendation.

Reviewer #2: All comments have been addressed

Reviewer #3: All comments have been addressed

2. Is the manuscript technically sound, and do the data support the conclusions?

Reviewer #2: (No Response)

Reviewer #3: Yes

3. Has the statistical analysis been performed appropriately and rigorously? 

Reviewer #2: (No Response)

Reviewer #3: Yes

4. Have the authors made all data underlying the findings in their manuscript fully available?

Reviewer #2: (No Response)

Reviewer #3: Yes

5. Is the manuscript presented in an intelligible fashion and written in standard English?

Reviewer #2: (No Response)

Reviewer #3: Yes

6. Review Comments to the Author

Reviewer #2: Just one remaining suggestion to the authors: adjust the abstract to clarify that the paper tests the specific mechanisms that could generate the winner-loser gap, rather than testing the gap itself

Reviewer #3: I have carefully reviewed the revised version of the manuscript and am pleased to see that the authors have addressed all of the suggestions I previously provided. Moreover, they have thoughtfully incorporated feedback from other reviewers, which has significantly strengthened the paper. The revisions have enhanced both the clarity and the overall contribution of the study, making it more compelling and interesting. I have no further comments or suggestions at this time.

7. PLOS authors have the option to publish the peer review history of their article (what does this mean?). If published, this will include your full peer review and any attached files.

Reviewer #2: **Yes: **Sergio Simoni Jr.

Reviewer #3: No

---

## [Editor Report · Acceptance letter]

6 Dec 2024

PONE-D-24-06817R1 

PLOS ONE

Dear Dr. Nemcok, 

I'm pleased to inform you that your manuscript has been deemed suitable for publication in PLOS ONE. Congratulations! Your manuscript is now being handed over to our production team.

Kind regards, 

on behalf of

Dr. Ivan Filipe de Almeida Lopes Fernandes 

Academic Editor

PLOS ONE